# Responses of Plant Bud Bank Characteristics to the Enclosure in Different Desertified Grasslands on the Tibetan Plateau

**DOI:** 10.3390/plants10010141

**Published:** 2021-01-12

**Authors:** Xinjing Ding, Peixi Su, Zijuan Zhou, Rui Shi, Jianping Yang

**Affiliations:** 1Key Laboratory of Land Surface Process and Climate Change in Cold and Arid Regions, Northwest Institute of Eco-Environment and Resources, Chinese Academy of Sciences, 320, Donggang West Road, Lanzhou 730000, China; dingxj@lzb.ac.cn (X.D.); zhouzzj@lzb.ac.cn (Z.Z.); shirui@lzb.ac.cn (R.S.); yangjianping20@nieer.ac.cn (J.Y.); 2Chinese Academy of Sciences, 19A Yuquan Road, Beijing 100049, China

**Keywords:** Tibetan Plateau, desertified grassland, grassland enclosure, bud bank, meristem limitation index

## Abstract

Asexual reproduction is the main mode of alpine plant reproduction, and buds play an important role in plant community succession. The purpose of this study is to explore whether the desertified grassland can recover itself through the existing bud bank. The bud bank composition, distribution and size of different desertified grasslands were studied using unit volume excavation on the Tibetan Plateau. The bud bank consisted of tiller, long and short rhizome buds, and more than 40% of buds were distributed in the 0–10 cm soil layer. Enclosure changed the bud density, distribution and composition. The bud densities were 4327 and 2681 No./m^2^ in light and middle desertified grasslands before enclosure, while that decreased to 3833 and 2567 No./m^2^ after enclosure. Tiller bud density and proportion of middle desertified grassland were the highest, increased from 2765 (31.26%, before enclosure) to 5556 No./m^3^ (62.67%, after enclosure). There were new grasses growing out in the extreme desertified grassland after enclosure. The meristem limitation index of moderate desertified grassland was the lowest (0.37), indicating that plant renewal was limited by bud bank. Plants constantly adjust the bud bank composition, distribution, and asexual reproduction strategy, and desertified grasslands can recover naturally, relying on their bud banks through an enclosure.

## 1. Introduction

Grassland is an important component of the earth’s terrestrial ecosystems, as it accounts for 40% of the total land area; therefore, the stability of grassland ecosystems has an important ecological and economic implications [1]. Grassland desertification has occurred to varying degrees worldwide as a consequence of anthropogenic (e.g., overgrazing) and natural factors (e.g., climate change) [2]. Grassland desertification results in reductions of grass shoot density and biomass, increases of the poisonous weeds, decreases of livestock capacity, and deterioration of the soils physical and chemical properties [3]. In addition, grassland desertification can also lead to a series of other serious problems, such as soil erosion, drought, and the transmission of rodent and insect diseases [4,5]. Thus, the restoration of desertified grassland has become one of the most urgent ecological and environmental problems [6,7].

Research on the desertified grassland restoration and management began with the study of the relationship between grazing and vegetation reconstruction in the tallgrass prairies of North America in the 1930s [8]. Subsequent breakthroughs include the development of grassland plant reconstruction technology and the grassland rebuilding [9,10]. Currently, research on grassland desertification were primarily concentrated on arid and semi-arid areas, whereas research on alpine grasslands mostly focused on the restoration of degraded grassland [11]. However, research on the process, characteristics and restoration of alpine grassland desertification has been relatively understudied. Low temperatures and abundant precipitation make alpine grasslands fundamentally different from grasslands in arid and semi-arid areas; therefore, more targeted research is needed for the ecological management of alpine grassland desertification.

The Zoige Plateau in the eastern Tibetan Plateau is one of the most important plateau wetlands in the world, because it is not only a key reservoir of plateau biodiversity and the center of differentiation of the world’s mountain biological species but also the main collection area for runoff of the Yellow River and an important water conservation area [12,13]. The average elevation of this region is 3440 m, and the vegetation is seminatural due to grazing (the grassland rotates grazing in winter and summer). The plants are mainly grass and sedge, and *Elymus nutans* and *Kobresia setchwanensis* are the dominant plants [6,13]. Currently, nearly 90% of alpine grasslands on the Yellow River basin show varying degrees of desertification. Desertification of these grasslands result in increased sediment content in the Yellow River and thus directly affects the ecological security and economic development of the basin. At present, desertified grasslands are divided into different grades according to vegetation coverage and soil characteristics, and different measures are taken to restore desertified grasslands [14,15]. Given that the threat posed by alpine grassland desertification is becoming increasingly serious, economical and effective means of restraint alpine grassland desertification and grassland restoration has become a major focus [16].

The collection of all of the buds potentially available for plant asexual propagation is called a bud bank [17,18]. A bud bank is a key part of plant adaptation to environmental stress and vegetation restoration, as it directly affects the resilience of degraded ecosystems. The propagules (i.e., seeds and buds) provide both a simple and effective means for plants to recover degraded ecosystems without destroying genetic diversity [19]. In alpine areas, low temperatures and short growth periods are not conducive to the seed maturing. In addition, because livestock selectively feeds on grass, overgrazing exacerbates the phenomenon, which restrains grass growth and seed production. Plants cannot carry out community renewal and succession through sexual reproduction, so asexual reproduction becomes the main reproductive approach [20,21,22,23,24]. For example, 99% of the shoots in North American grasslands are formed by bud banks [25], and 84% of grasslands in the Alps are occupied by cloned plants [26]. In grassland ecosystems, the size and composition of the bud bank reflect the plant richness and composition [21,27,28]. In some ecosystems, the bud bank determines the community dynamic under drought, livestock feeding or invasion of alien species, and bud bank affects community structure and productivity [29,30]. Since 1977, studies of belowground bud bank have been conducted in various habitats worldwide, and these studies have enhanced our understanding of bud banks and their implications on plant communities [17,21,31], such as grazing [32], fire [33], precipitation [25,34] and soil nutrition [35,36], aeolian disturbance [37]. Currently, our understanding of bud banks in alpine areas is relatively poor, especially the effect of grassland desertification on bud banks.

The main purpose of this study was to understand the characteristics of the belowground bud bank of alpine desertified grassland, explore whether grasslands with different degrees of desertification can be restored naturally by relying on the existing belowground bud bank. For this purpose, we propose two hypotheses. First, with the grassland desertification degree increasing, bud and shoot density decrease gradually, grassland enclosure eliminates the interference of cattle and sheep on plants (feeding and trampling) and improves the bud and shoot density of different desertified grasslands, and then increases the grassland productivity. Second, the enclosure change bud bank composition and distribution in the different desertified grasslands, the proportion of tiller bud increases, while the proportion of rhizome bud decreases, and the asexual reproduction strategy of different desertified grassland plants change from an expansive to conservative strategy. This study will expand our understanding of the ecological restoration mechanism of alpine desertified grasslands under enclosed conditions and provides scientific information on the restoration and management of desertified grasslands.

## 2. Materials and Methods

### 2.1. Site Description

The study area was located in the Zoige Plateau in the eastern Tibetan Plateau, China. The Zoige Plateau is affected by the southeast and southwest monsoon, has a semi-humid continental monsoon climate in the cold temperate zone of the plateau and is sensitive to climate change. Its climate is cold and humid, with a long winter and no summer. The annual average temperature is 1.1 °C, and the annual rainfall is 615 mm [38,39,40]. The altitude of the study area is 3440 m, and the geographical coordinates are 33°55′ N, 102°09′ E. According to the vegetation coverage and quicksand area, desertified grasslands were divided into light desertified grassland (LD), moderate desertified grassland (MD), severe desertified grassland (SD) and extreme desertified grassland (ED) (Figure 1, Table 1) [14,15]. From September 2018, grassland enclosures were conducted for each desertified grassland.

### 2.2. Sampling Method

The belowground bud bank reached its maximum and stable value at the end of the growing season [34]; therefore, we investigated the plant communities and sampled their bud bank in each desertified grassland in late September of 2018 and 2019. The area of each desertified grassland exceeds 5 ha, the distance between each desertified grassland is more than 200 m. Three 1 × 1 m^2^ plots were randomly set up on each desertified grassland to investigate vegetation composition, coverage, height and abundance, and the spacing among plots was more than 50 m. Plants with taxonomic uncertainty followed the nomenclature of the flora of the China online database (http://www.iplant.cn/frps). The plant composition is shown in Table 2 [41]. The aboveground parts of the plant in the plots were harvested and stored in paper bags. The samples were brought back to the laboratory and dried to constant weight at a constant temperature of 80 °C. The aboveground biomass were weighed by Huaxing Electronic Balance (accurate to 0.01 g), and the aboveground biomasses were then counted.

We threw a ring on the grasslands, and the spots where the ring landed were the sampling spots. Three samples were sampled from each desertified grassland before and after enclosure, and a total of 24 samples were obtained. Each sample was 15 × 15 × 30 cm in length, width and depth, and the depth was divided into three layers: surface (0–10 cm), middle (10–20 cm) and lower (20–30 cm). The bud bank samples were placed in ice bags and brought back to the laboratory. Each sample was washed with water, and the buds were counted by soil layer and bud type [34].

### 2.3. Experimental Method

In this study, only live buds were counted, and dead buds were manually removed. The live and dead buds could be distinguished based on their morphology and color. The types of vegetative offspring were determined based on their morphology. Three types of buds were distinguished: tiller bud, long and short rhizome bud (Figure 2). Because of differences in bud morphology, the different types of buds were counted in different ways: the long and short rhizome buds were directly counted, whereas the tiller buds needed to be dissected at the base of the plant and counted. Short rhizome buds exist in the vertical direction, while long rhizome buds exist in the horizontal direction. After counting plant buds, the belowground parts were dried at a constant temperature (80 °C), and the belowground biomasses were quantified [34].

### 2.4. Statistical Analyses

The ratio of bud density to shoot density (bud density/shoot density) is the plant meristem limitation index, which indicates whether the replacement of the aboveground plant is limited by the bud bank [25].

The original data of bud, shoot and biomass densities were converted into numbers per square meter, and then the average value of each sampling position was calculated for further analysis. One-way ANOVA was applied to analyze differences in the belowground bud and biomass densities of the grasslands before and after enclosure. The significance threshold was *p* < 0.05, and the extremely significant threshold was *p* < 0.01. The data were processed by SPSS 22.0, and figures were drawn with Origin graphing 2017 software.

## 3. Results

### 3.1. The Effect of Enclosure on the Bud Bank Distribution

The bud density of LD was the highest while that of ED was the lowest (Figure 3). The enclosure reduced the bud density of LD and MD, and the bud density was 4237 and 2681 No./m^3^ before enclosure, while after enclosure the buds were 3833 and 2567 No./m^3^, respectively, with a decrease of 9.53% and 4.28%. The bud density of ED increased from 0 (before enclosure) to 433 No./m^2^ (after enclosure). More than 40% of the buds were distributed in the 0–10 cm layer, and enclosure altered the distribution of bud banks (Figure 4). Grassland enclosure increased the proportion of 0–10 cm buds of LD and MD from 41.26% and 57.46% (before enclosure) to 46.95% and 74.03% (after enclosure). In contrast, the proportion of 0–10 cm buds of the SD decreased from 71.60% (before enclosure) to 43.24% (after enclosure).

### 3.2. The Effect of Enclosure on the Bud Bank Composition

The bud bank of desertified grasslands primarily consisted of short rhizome buds (25.3–83.3%), followed by tiller buds (Figure 5). Grassland enclosure changed the bud bank composition; specifically, the enclosure increased the density and proportion of tiller buds. The density and proportion of tiller buds in MD were the highest, 2765 No./m^3^ (31.26%, before enclosure) and increased to 5556 No./m^3^ (62.67%, after enclosure). Tiller bud density in ED was 0 No./m^3^ before enclosure, while after enclosure, it increased to 222 No./m^3^, accounting for 12.67% of the total buds. Before enclosure the short rhizome buds proportion of total buds in LD and MD were 83.3% and 53.62%, and after enclosure, it decreased to 68.33% and 25.33%.

### 3.3. The Effect of Enclosure on Plant Species and Aboveground Biomass

Plant species and aboveground biomass were the highest in LD and the lowest in ED (Table 3, Figure 6). Generally, grassland enclosure increased the plant species and aboveground biomass. The aboveground biomass of LD increased by 10.03% to 186.91 g/m^2^ after enclosure, and the aboveground biomass of MD was second to that of LD, which increased by 17.04 g/m^2^ after enclosure. After the enclosure, there were plants in the ED and aboveground biomass was 33.3 g/m^2^.

### 3.4. Meristem Limitation Index along Desertification Gradient

If the meristem limitation index is less than one, or only slightly larger than one, then the replacement of the aboveground plant is limited by the bud bank. If the index is much greater than one, then the replacement of the aboveground plant is not restricted by the bud bank. The meristem limitation indexes of LD, SD and ED were greater than one. The meristem limitation index was the highest (13.54) in ED and the lowest (0.37) in MD (Table 4), indicating that the replacement of the aboveground plant in MD is restricted by the bud bank.

## 4. Discussion

In alpine desertified grasslands, more than 40% of the buds were distributed in the 0–10 cm soil layer; while the research of Zhang et al. showed that belowground buds were primarily distributed in the 10–30 cm soil layer in Mu Us sand land (which has a temperate continental climate, average elevation is 1430 m), there are some differences in the bud bank distribution between sand land and alpine desertified grasslands [3]. Plants alter the distribution of bud banks to adapt to environmental changes [21], the environment contributes to determining the distribution of bud banks. In sandy land, plant rhizomes in the 0–10 cm soil layer are vulnerable to high temperature and drought stress; consequently, bud survival and sprouting rate are low [3]. While buds in the 10–30 cm soil layer are protected from high temperature and drought stress, the rate of sprouting and growing out of the soil are relatively high. However, the sprouting cost of bud in the alpine grassland is different from sandy land, which is related to the relative positions of buds in the soil. The sprouting cost of buds deep in the soil is much higher than that of shallow ones [42,43,44]. For this reason, alpine grassland buds were primarily distributed in the 0–10 cm soil layer, which was conducive to the sprouting and growth of plants.

The energy storage and expansion capabilities of the bud differ as a consequence of the morphology of buds attached to organs [42,43,45]. Therefore, different bud banks permit plants to employ either an expansive or conservative strategy to respond to environmental disturbances and the availability of resources [18,46,47], some plants even employ both expansive and conservative strategies. Before enclosure, the grassland was disturbed by overgrazing, and the plants adopted the expansive strategy, which could make the daughter plants escape the grazing and trampling of cattle and sheep. Owing to long-term overgrazing, shoot density of desertified grassland—especially in SD and ED—was low. The enclosure eliminates interference from cattle and sheep and increased the proportion of tiller buds. In such a scenario, plants adopt a conservative strategy: the increased amount of tiller buds around the mother plant, which quickly sprout as daughter plants. This strategy allows plants to effectively take advantage of the available resources, expand their population and occupy the existing habitat to resist the invasion of alien plants [48], which also confirms our second hypothesis. Grassland plants continue to alter the bud composition depending on the degree of desertification and grassland enclosure, indicating that the alpine belowground bud bank maintains a certain degree of plasticity in the spatial structure, as the two strategies can be variously employed to cope with environmental stochasticity [49].

The meristem limitation index characterizes the extent to which the plant shoot replacement restricted by bud banks. The meristem limitation index was 4.67 and 13.54 in SD and ED, respectively, greater than one. To fully occupy the existing habitat, more belowground buds were generated, resulting in a meristem limitation index greater than one for SD and ED [23]. Long-term overgrazing resulted plants to be eaten by livestock in SD and ED, plant shoots were sparse and the bud bank size was small, and there were a lot of blank habitats. At the moment, plants do not fully occupy the grassland, and grassland renewal is still limited by the bud bank. This also verified the applicability of the plant meristem limitation index, suggesting that the plant meristem limitation index may only be applicable to stable plant communities and not to the recovering desertified grasslands. Although the plant shoot density of MD was the highest, the meristem limitation index was 0.37, indicating that the plants of MD were relatively scarce and did not fully occupy the existing habitat. Thus, the productivity of MD was still limited by the bud bank size, it also indicates that it will take a long time for the enclosure restoration of desertified grasslands [23]. Therefore, artificial measures such as breeding could be taken to increase the plant propagules size, speed up the process of grassland restoration and shorten the recovery time.

In this study, aboveground biomass and the bud density increased gradually with the desertification decrease of the alpine grasslands. Many studies have shown that when there are not enough buds to replenish the aboveground plants in time, plant sprouting may be limited by the bud bank size, which may inhibit the primary productivity of the ecosystem [25,50]. Studies have showed that the annual variation of net primary productivity of grasslands are small [27,51]. The main reason for this small difference might stem from the small bud bank size in arid areas [27,51]. Specifically, when precipitation meets the demands for plant growth, the belowground buds are not sufficient to supplement aboveground plants, thereby resulting in no significant change of productivity in the abundant precipitation season [27,51,52]. Therefore, the bud bank is an important factor restricting grassland productivity in arid areas [27,51]. From ED to MD, plant bud density and aboveground biomass both showed a gradually increasing trend. As bud density increased, plant bud banks gradually met the growth needs; consequently, bud density, shoot density and aboveground biomass showed consistent increase trends. However, the shoot density of LD was lower than that of MD, but the aboveground biomass of LD was higher than that of MD. This indicates that grassland productivity is not only limited by the bud bank, and it also verifies that the first hypothesis in this paper is not completely correct [27,51,52].

## 5. Conclusions

The bud bank of desertified grassland is composed of tiller, long and short rhizome bud, and short rhizome bud is dominant, accounting for 25–83%. More than 40% of the bud bank are distributed in the 0–10 cm soil layer, and plants adopt a conservative strategy to rapidly reproduce to occupy the habitat. Desertified grasslands can recover via the existing bud banks depending on the enclosure. The bud bank size of extreme desertified grassland is small, and artificial breeding could shorten the recovery time.

## Figures and Tables

**Figure 1 plants-10-00141-f001:**
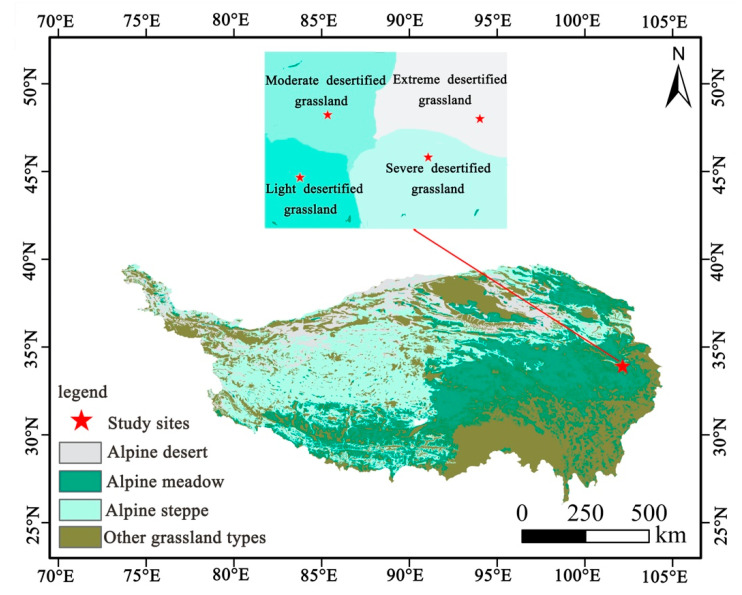
Distribution of study sites on the Tibetan Plateau.

**Figure 2 plants-10-00141-f002:**
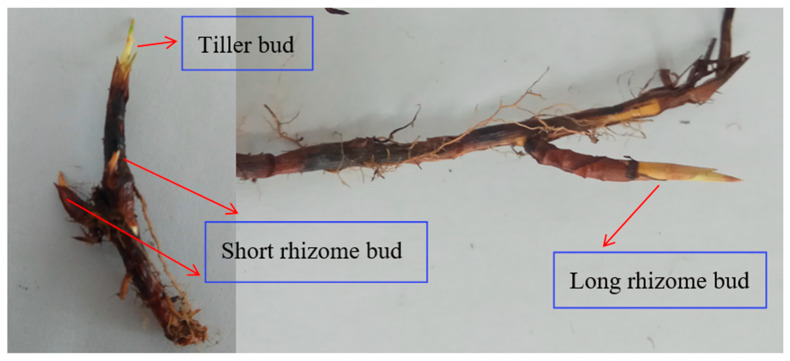
The three investigated bud types.

**Figure 3 plants-10-00141-f003:**
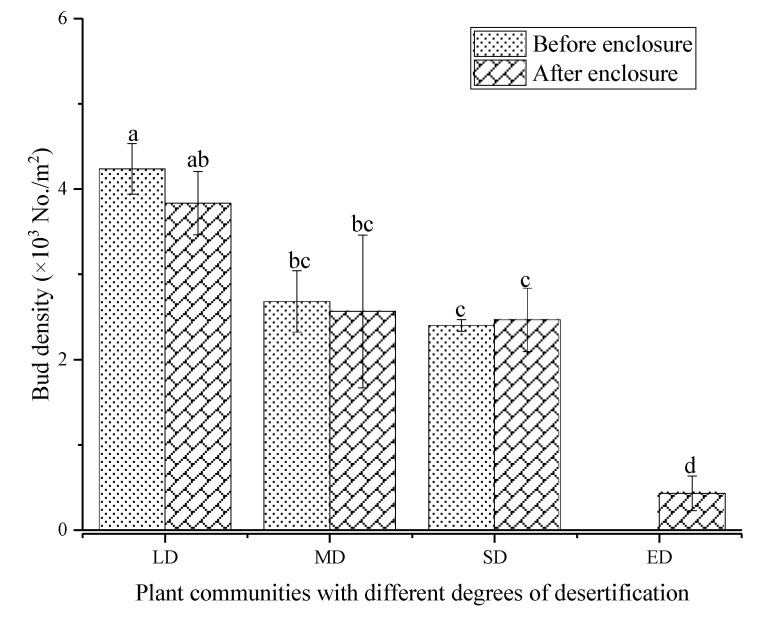
Variation characteristics of bud banks in different desertified grasslands. LD: light desertified grassland; MD: moderate desertified grassland; SD: severe desertified grassland; and ED: extreme desertified grassland. Data with different letters in the same column indicate significant differences at the 0.05 level.

**Figure 4 plants-10-00141-f004:**
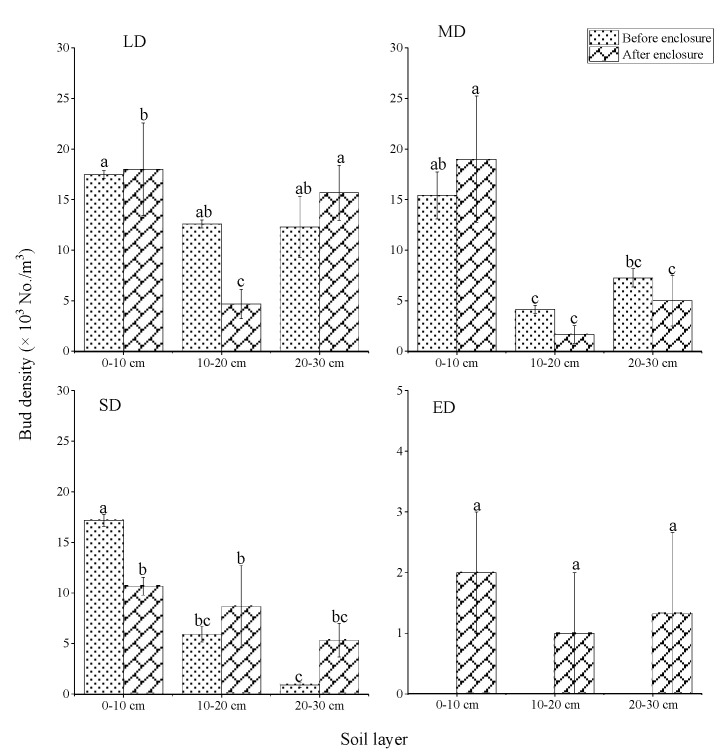
Variation characteristics of bud bank distribution in different soil layers. LD: light desertified grassland; MD: moderate desertified grassland; SD: severe desertified grassland; and ED: extreme desertified grassland. Data with different letters in the same column indicate significant differences at the 0.05 level.

**Figure 5 plants-10-00141-f005:**
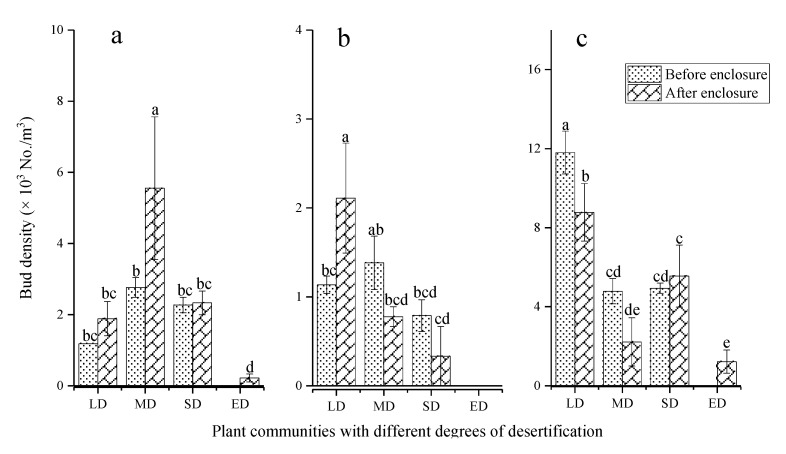
Variation characteristics of different bud densities in desertified grasslands. (**a**): tiller buds, (**b**): long rhizome buds, (**c**): short rhizome buds. LD: light desertified grassland; MD: moderate desertified grassland; SD: severe desertified grassland; and ED: extreme desertified grassland. Data with different letters in the same column indicate significant differences at the 0.05 level.

**Figure 6 plants-10-00141-f006:**
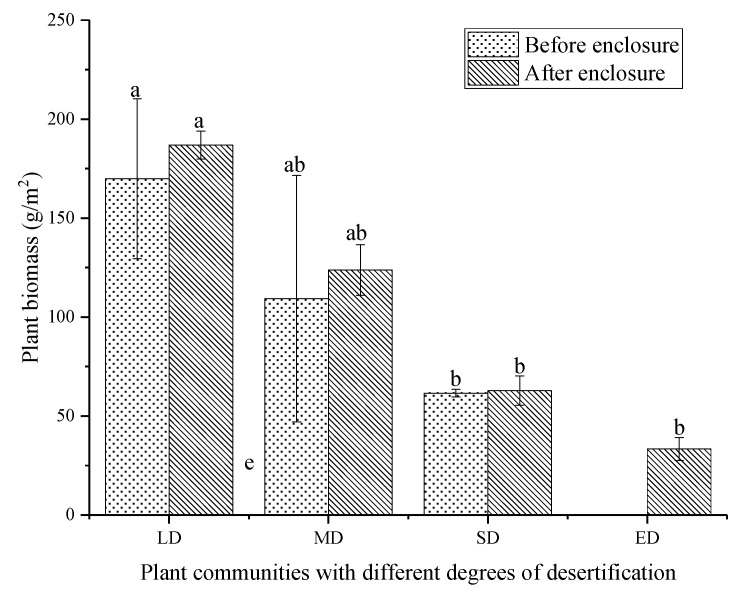
Aboveground biomass of different desertified grasslands. LD: light desertified grassland; MD: moderate desertified grassland; SD: severe desertified grassland; and ED: extreme desertified grassland. Data with different letters in the same column indicate significant differences at the 0.05 level.

**Table 1 plants-10-00141-t001:** Classification and characteristics of desertified grasslands.

Community Type	Vegetation Coverage/%	Quicksand Area/%
LD	50 ≤ C < 60	5 ≤ S < 10
MD	20 ≤ C < 50	10 ≤ S < 30
SD	10 ≤ C < 20	30 ≤ S < 50
ED	<10	≥50

Note: LD: light desertified grassland; MD: moderate desertified grassland; SD: severe desertified grassland; ED: extreme desertified grassland.

**Table 2 plants-10-00141-t002:** Plant species and coverage of different desertification grasslands.

Community Type	Plant Species	Coverage %
Light desertified grassland	*Elymus nutans*	23.3
	*Leymus secalinus*	18
	*Anaphalis lactea*	15
	*Poa pratensis*	6.6
	*Ligularia virgaurea*	3
	*Artemisia hedinii*	2.8
	*Agropyron cristatum*	2.7
	*Lancea tibetica*	2.7
	*Kobresia setchwanensis*	1
	*Potentilla anserina*	0.2
Moderate desertified grassland	*Kobresia setchwanensis*	21.3
	*Anaphalis lactea*	12.3
	*Potentilla anserina*	2.3
	*Stellera chamaejasme*	1.1
	*Leymus secalinus*	0.9
	*Ajuga lupulina*	0.4
Severe desertified grassland	*Carex moorcroftii*	5.3
	*Leymus secalinus*	4.9
	*Elymus nutans*	1.9
	*Dracocephalum heterophyllum*	1.2
	*Gentiana straminea*	0.8
Extreme desertified grassland	*Agropyron cristatum*	4.8
	*Chenopodium aristatum*	0.3

**Table 3 plants-10-00141-t003:** Plant species in different desertified grasslands.

Community Type	Plant Species	
	Before Enclosure	After Enclosure
LD	3 ± 1 a	4 ± 0.6 a
MD	2.7 ± 0.3 a	3 ± 0 ab
SD	2 ± 0 a	2 ± 0 b
ED	-	1.3 ± 0.3 c

LD: light desertified grassland; MD: moderate desertified grassland; SD: severe desertified grassland; and ED: extreme desertified grassland. Data with different letters in the same column indicate significant differences at the 0.05 level.

**Table 4 plants-10-00141-t004:** Shoot density and the meristem limitation index of desertified grasslands.

Plant Community	Shoot Density Number/m^2^	Meristem Limitation Index
LD	3589 ± 3030 a	1.07 ± 0.24 c
MD	6891 ± 2589 a	0.37 ± 0.06 d
SD	528 ± 25 a	4.67 ± 0.67 b
ED	32 ± 16 a	13.54 ± 1.36 a

LD: light desertified grassland; MD: moderate desertified grassland; SD: severe desertified grassland; and ED: extreme desertified grassland. Data with different letters in the same column indicate significant differences at the 0.05 level.

## Data Availability

All data reported here is available from the authors upon request.

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
