# Peer review of "Responses of Plant Bud Bank Characteristics to the Enclosure in Different Desertified Grasslands on the Tibetan Plateau"

_plants, 2021, doi:10.3390/plants10010141_

Round 1

Reviewer 1 Report

Report for Plants Manuscript ID plants-1041776...

Title ‘Responses of plant communities and bud bank characteristics to enclosure in different desertified grasslands on the Tibetan Plateau ‘

English language and style are fine/minor spell check required

General comments: -

In general, the topic is very interesting and has a great impact. I found the paper to be overall well prepared. I found the manuscript is well organized, but I have some minor comments.

Detailed comments: - 

Title Please

change enclosure to ....the enclosure                                                                      

Abstract

This section is missing the direct aim of the study. Please state the aim of the study clear in this section.

Line 13: please add the word propagation to the first sentence to become the main mode of plant propagation

  • Please put all the most significant data to the abstract

Introduction

Line 83: please change the word deepening to depth

Line 90: Please change further our understanding to… expand our understanding

Line 90: please change the word bud to buds for verb-subject match

Results

Abbreviations (LD, ED, MD, and SD) First time mentioned, need to be written in a complete form and write the abbreviation for each one in parentheses beside it

Figure 1

Line 106-108 please adjust English style and formatting Also, Change the sentence; Data with different letters indicate significant differences at the 0.05 level in the same column to Data with different letters in the same column indicate significant differences at the 0.05 level. Please apply this rule to the entire manuscript in the places that it is mentioned

Line 135 please change and lowest to and the lowest

Materials and Methods

Line 243-244: please change    origin 2017 to origin graphing 2017 software

Conclusions

Line 111: 2.2 changes in the the bud bank …(please take off the extra the)

Line 135: please change were variational to…. were varied

This section is too general. please include the most significant data with a suitable recommendation.

Reviewer 2 Report

The manuscript of Ding et al. is a well written manuscript, focused on it’s topic, understandable in the first read. The manuscript focuses on the desertification and grazing enclosure, but contrary to the studies of similar topic, they focus on alpine regions. Having a desertification gradient, they are able to determine in which case is the bud bank able to support aboveground revegetation in case of grazing enclosure. After a second read I found some mistakes and I have some comments that could make the manuscript even better. Overall I support the publication of this work, as it is a good and interesting work.

The Abstract needs to be revised. I found missing words and repetitions, as mentioned below:

„Asexual reproduction is the main mode of plant ???”

Starting from line 19, for me it seems more understandable in this form:

Alpine grasslands bud bank consisted of tiller, long and short rhizome buds. Grassland enclosure altered bud bank composition and increased the tiller bud density and proportion. Grassland enclosure reduced the bud density of light and middle desertified grasslands, improved plant diversity and aboveground biomass, and plants began to appear in the extreme desertified grassland after enclosure. The plant meristem limitation index of extreme desertified grassland was the highest (13.54) and that of moderate desertified grassland was the lowest (0.37). Thus, grassland plants can adjust their reproduction strategies based on environmental changes, and the natural recovery of desertified grasslands could be achieved by grassland enclosure.

P2 L83. Please specify: the „deepening of desertified grassland” means the increasing soil depth or increasing desertification level. Also, the first hypothesis can be separated into two, one referring to the dud bank of soil depth/desertification level, the other one to the effect of enclosure.

P3 L94. As your main questions focus on desertification and enclosure, I suggest to use new subtitles (example: Bud bank changes along desertification gradient, Effect of enclosure to bud bank) to highlight their effect. The text of the manuscript should also be changed to correspond with the subtitles.

P7 L165. „alter” instead of „altered”

P7 L173. Not correct: „was size”. Besides, I would replace the sentence „to fully occupy…” after the part where you present the values. This would highlight, that bud bank really increases due to enclosure, which is good, but the effect of overgrazing was so negative, that this increase is still not enough.

P7 L176. So whatever the meristem limitation index is (over or bellow 1), bud bank it is not enough to support grassland revegetation?

P8 L204. The categorization in based of literature or on personal observations? Also, you should mention how big was the area where enclosure was applied.

P8 L215. Why do you choose to have so low number of replications and big distances between the plots?

Please also cite the literature based on what you determined your plant species.

Please, also indicate the area from which the biomass samples were collected and how were they stored, how were they dried and measured (accuracy, name and type of the balance).

P9 L224.You mean one soil sample/plot was collected? Conform the methodology used to sample bud bank, it is enough one sample/plot to adequately measure existing bud bank?

P9 L236. Please correct: „are exist”. Also, it would be nice if you could provide pictures of different sampling areas and bud types.

P9 L239. Did you calculated a mean for the samples of the two sampling years or how did you handled the observations coming from the two different years?

Reviewer 3 Report

The article ‘Responses of plant communities and bud bank characteristics to enclosure in different desertified grasslands on the Tibetan Plateau’ concern on the interesting problem of belowground bud bank in alpine grassland in differed desertification stages presenting the outcoming climate changes.

The abstract should be more informative, and precise. It is a lack of justification for the study. The best arrangement of the abstract is dividing the text into parts: background, aim, approach, results and conclusions. In this abstract is lack of aims (What did you plan to achieve in this work? What gap is being filled?), and approach (what method was used, where was the study area placed?). Please rearrange the abstract in this way. The last sentence is very general and it is not a good conclusion of the study, please be more specific.

The introduction is well written, and concern on the most important issues. However it is a lack of alpine grasslands characteristic, e.g. are the alpine grasslands natural or seminatural vegetation? At what altitude does this type of vegetation occur? What are the dominant plant species? What type of management dominates? The aim of the study and hypothesis are well formulated.

The Material and methods chapter should be mover before the results.

The map of the area with site locations should be added, illustrating the distances among particular grasslands, and the altitude.

The theoretical background of dividing different types of grasslands should be added. Please add the references, and characteristic of the grasslands types. If it is Your own idea, and You have divided the types based on vegetation coverage, it should be described in more detail.

Table 4 presenting the species composition of particular grasslands types should be prepared more clearly. All species should be listening, not only common, and the coverage of each species should be added.

Presentation of the results should be changed. Your data show the state of the particular type of grassland, not the process of desertification, thus You should not use the terms: increase or decrease to compare the grasslands types. Please write in which type the value was lower or higher.

The description of the graphs should be more detail. Particularly in the case of figure 3 and 4. Please justify why do you show both of them.

Also, the results presented in Table 1 should be better described, and the table should be improved. Why do You show the results of different measures of diversity, like Species Richness, Simpson, Shannon-Wiener, and  Pielou indices? Maybe better it will be select one, the best showing the results.

Discussion is the best part of the article. However, some parts are simply a repetition of the Results part. I think, that it should be added to the practical implication of the study, as well as the limitation of the study.

I suggest also editing the text by a native speaker.

The detail comments are given directly in the text of the manuscript.

Reviewer 4 Report

Dear authors, I have read with interest your article.

You present an interesting and critical status for areas in one of the most important biomes in the world., also an interesting solution for its improvement.

There are some changes that I think need to be considered in order to improve your work.

Abstract

L13-14 Please rewrite the first sentence. It is hard to understand it. Especially main mode of plant - what do you refer.

L21 / L22 - you repeat Grassland enclosure2 times, and in line 19 your first word is Enclosure. Please change the sentences in Lines 21 and 22 in order to move Grassland enclosure in the sentence.

Results

Expand the results section.

L96-102. Add some more explanations regarding your findings. I think you have interesting results, and you should present them in a longer paragraph. 

Add to figure 2 a caption similar to figure 1 (LD: light
desertified grassland; MD: moderate desertified grassland; SD: severe desertified grassland; and ED: extreme desertified grassland. Data with different letters indicate significant differences at the 0.05 level in the same column.) - It will help to have standalone captions, and wil make your study easier to read and understand.

Similar to figure 2, add caption to figures 3 and 4.

L112-116. Add some more explanations regarding your findings. Expand the paragraph. It is too short for explaining 2 figures, each of them with 3 parts (a, b, c).

L124-127. Identic suggestion as above. Expand your results and explanations/interpretation.

L128 - Table note should be completed with LD: light
desertified grassland; MD: moderate desertified grassland; SD: severe desertified grassland; and ED: extreme desertified grassland.
Also, add same suplemmentary information to figure 5 (LD: light
desertified grassland; MD: moderate desertified grassland; SD: severe desertified grassland; and ED: extreme desertified grassland.)

L134-135 - Identic suggestion as above. Expand your results and explanations/interpretation.

L137-138 - Table 4. able note should be completed with LD: light desertified grassland; MD: moderate desertified grassland; SD: severe desertified grassland; and ED: extreme desertified grassland.

Discussion section. - This part of your work is interesting and well written.

Conclusions. Add some of your values in the conclusion section. Present your main findings and point them.

Overall, your article sounds good and promising. You need to better present your results in order to make them easier to understand, read and use as a base for future researches and other researchers.

Round 2

Reviewer 3 Report

The article is improved according my comments. I have only one suggestion; please improve the figure 1. The different types of grasslands listed in the frame are not shown on the figure. Thus the green frame with grasslands types is redundant.

Reviewer 4 Report

Dear authors, you have managed to improve a lot your work and sound better in this moment.

I hope that for the future this kind of approach will be largely used in grassland research.

Author Response

1. Dear authors, you have managed to improve a lot your work and sound better in this moment. I hope that for the future this kind of approach will be largely used in grassland research.

Response:

Many thanks to the reviewers for their comments and suggestions, which have gradually improved the quality of the article and the author's scientific research level. Thanks again to the reviewers and editor for their hard work.